# Effects of Acute and Chronic Gabapentin Treatment on Cardiovascular Function of Rats

**DOI:** 10.3390/cells12232705

**Published:** 2023-11-25

**Authors:** Ved Vasishtha Pendyala, Sarah Pribil, Victoria Schaal, Kanika Sharma, Sankarasubramanian Jagadesan, Li Yu, Vikas Kumar, Chittibabu Guda, Lie Gao

**Affiliations:** 1Department of Anesthesiology, University of Nebraska Medical Center (UNMC), Omaha, NE 68198, USA; vedpendyala27@gmail.com (V.V.P.); spribil@unmc.edu (S.P.); vicki.schaal@unmc.edu (V.S.); lyu@unmc.edu (L.Y.); 2Mass Spectrometry and Proteomics Core Facility, University of Nebraska Medical Center (UNMC), Omaha, NE 68198, USA; kasharma@unmc.edu (K.S.); vikas.kumar@unmc.edu (V.K.); 3Department of Genetics, Cell Biology and Anatomy, University of Nebraska Medical Center (UNMC), Omaha, NE 68198, USA; s.jagadesan@unmc.edu (S.J.); babu.guda@unmc.edu (C.G.); 4Center for Biomedical Informatics Research and Innovation, University of Nebraska Medical Center (UNMC), Omaha, NE 68198, USA

**Keywords:** gabapentin, arterial blood pressure, heart rate, left ventricular function, proteomics, bioinformatics, calmodulin

## Abstract

Gabapentin (GBP), a GABA analogue, is primarily used as an anticonvulsant for the treatment of partial seizures and neuropathic pain. Whereas a majority of the side effects are associated with the nervous system, emerging evidence suggests there is a high risk of heart diseases in patients taking GBP. In the present study, we first used a preclinical model of rats to investigate, firstly, the acute cardiovascular responses to GBP (bolus *i.v.* injection, 50 mg/kg) and secondly the effects of chronic GBP treatment (*i.p.* 100 mg/kg/day × 7 days) on cardiovascular function and the myocardial proteome. Under isoflurane anesthesia, rat blood pressure (BP), heart rate (HR), and left ventricular (LV) hemodynamics were measured using Millar pressure transducers. The LV myocardium and brain cortex were analyzed by proteomics, bioinformatics, and western blot to explore the molecular mechanisms underlying GBP-induced cardiac dysfunction. In the first experiment, we found that *i.v.* GBP significantly decreased BP, HR, maximal LV pressure, and maximal and minimal dP/dt, whereas it increased IRP-AdP/dt, Tau, systolic, diastolic, and cycle durations (* *p* < 0.05 and ** *p* < 0.01 vs. baseline; *n* = 4). In the second experiment, we found that chronic GBP treatment resulted in hypotension, bradycardia, and LV systolic dysfunction, with no change in plasma norepinephrine. In the myocardium, we identified 109 differentially expressed proteins involved in calcium pathways, cholesterol metabolism, and galactose metabolism. Notably, we found that calmodulin, a key protein of intracellular calcium signaling, was significantly upregulated by GBP in the heart but not in the brain. In summary, we found that acute and chronic GBP treatments suppressed cardiovascular function in rats, which is attributed to abnormal calcium signaling in cardiomyocytes. These data reveal a novel side effect of GBP independent of the nervous system, providing important translational evidence to suggest that GBP can evoke adverse cardiovascular events by depression of myocardial function.

## 1. Introduction

Gabapentin (GBP) is a 3,3-disubstituted derivative of gamma-aminobutyric acid (GABA). It is recommended as a first-line treatment for chronic neuropathic pain, particularly in diabetic neuropathy and post-herpetic neuralgia [1], and as an anticonvulsant to help control partial seizures in the treatment of epilepsy [2]. GBP is also used to treat an array of other central nervous system-associated disorders, including insomnia, drug and alcohol addiction, anxiety, bipolar disorder, borderline personality disorder, menopausal conditions, vertigo, pruritic disorders, and migraines [3]. However, increasing evidence indicates that GBP is misused and abused, with various side effects such as central hypoventilation, deficits in visual field, suicidal behavior, somnolence, dizziness, mood changes, and tiredness [4].

Although most GBP therapeutic side effects are evoked by alterations in neural function, clinical and animal studies increasingly suggest there are non-neural effects that directly act on peripheral tissues and organs, particularly the cardiovascular system. Designed as an analog of GABA, GBP binds with high affinity to the α2δ subunits of voltage-gated Ca^2+^ channels (VGCCs) [5] and reduces the high-threshold Ca^2+^ currents of presynaptic membranes. This leads to the inhibition of the release of the excitatory amino acids glutamate and aspartate [1]. Since the α2δ subunits are expressed in all subtypes of VGCCs [6], which are present not only in neurons but also in all other excitable cells (including skeletal muscle, vascular smooth muscle, and ventricular myocytes) [7], the influence that GBP has on the functioning of non-nervous systems and its relevant side effects should not be ignored.

Indeed, a retrospective cohort study revealed a significantly increased risk of adverse cardiovascular events, including heart failure, myocardial infarction, peripheral vascular disease, stroke, deep venous thrombosis, and pulmonary embolism in patients with diabetic neuropathy following long-term use of GBP [8]. Furthermore, several clinical cases of GBP-induced atrial fibrillation [9], acute heart failure [10,11], cardiomyopathy [12], and intraoperative hypotension [13] have been reported recently. Thus, we postulate that these events may be attributed, in part, to the effects of GBP on Ca^2+^ currents in cardiomyocytes or vascular smooth muscles.

Evidence from animal studies suggest that GBP can impact cardiovascular function by both central and peripheral mechanisms. Chen et al. [14] found that microinjecting GBP into the nucleus tractus solitarii (NTS) of the brainstem induced prominent dose-related depressor and bradycardic responses in spontaneously hypertensive rats (SHR) via nitric oxide (NO)-dependent mechanisms. Behuliak et al. [15] demonstrated that acute intravenous injections of GBP lowered blood pressure and heart rate by a sympatho-inhibitory mechanism in both SHR and normotensive rats (WKY), with more pronounced effects observed in the former. Interestingly, by utilizing pressure arteriography and whole-cell voltage-clamp techniques, Largeau et al. [16] found that GBP dramatically decreased the myogenic tone of isolated mesenteric arteries and evoked weak vasodilation in endothelium-denuded vessels, but it had no significant effect on Ca_v_1.2 currents in ventricular cardiomyocytes. These studies suggest that GBP-induced peripheral edema and acute heart failure may be due to the effects of GBP on heart and blood vessels. However, to the best of our knowledge, no studies have investigated GBP’s influence on left ventricular (LV) function in intact animals. In the present study, we evaluated the effects of acute and chronic administration of GBP on hemodynamics in isoflurane-anesthetized rats to address the hypothesis that GBP can suppress cardiac function. In addition, we performed label-free mass spectrometry-based proteomics of the LV myocardium followed by bioinformatic analysis to explore the molecular mechanisms and intracellular pathways underlying the GBP effects on cardiomyocytes.

## 2. Materials and Methods

All animal procedures were conducted in accordance with the guidelines of the National Institutes of Health Guide for the Care and Use of Laboratory Animals and conformed to ARRIVE Guidelines (https://www.nc3rs.org.uk/arrive-guidelines, accessed on 9 May 2023), as approved by the Animal Care and Use Committee of the University of Nebraska Medical Centre (UNMC-IACUC Protocol #17-080-09).

Eighteen Fisher 344 rats (3–4 months old), which included both sexes, were assigned to two groups. Group 1 (4 rats) was used to determine the acute cardiovascular responses to a bolus intravenous injection (*i.v.*) of GBP (50 mg/kg; Gabapentin, PHR1049-1G, Sigma-Aldrich, St. Louis, MO, USA). Group 2 (10 rats) was used to determine the effects of 7 days of intraperitoneal injections (*i.p.*) of GBP (100 mg/kg) on cardiovascular function and, in addition, determine the proteomic profile of the myocardium.

### 2.1. Arterial Blood Pressure (AP), Heart Rate (HR), and Left Ventricular (LV) Hemodynamics Measurements

Rats were anesthetized with 3% isoflurane delivered in 100% oxygen, lying on an electric heating pad to maintain body temperature at 37 °C. A PE10 catheter was inserted into the superior vena cava through the right jugular for vein saline supplementation and *i.v.* injection of GBP. A catheter-tipped transducer (SPR-407, Millar Instruments, Houston, TX, USA) was advanced via the right femoral artery into the abdominal aorta to measure BP and HR, whereas another catheter-tipped transducer (SPR-320NR, Millar Instruments, Houston, TX, USA) was advanced via the right carotid artery into the LV chamber to measure cardiac function. The transducer signals were input into a PowerLab data-acquisition system with sampling rate at 2 k/s, from which the data were monitored, recorded, and saved in a computer by the LabChart 8 software and analyzed using the Blood Pressure Module software v1 (AD Instruments, Colorado Springs, CO, USA) with an average of 50–100 cycles.

### 2.2. Plasma Norepinephrine Assay

A volume of 300 μL of plasma was used to measure norepinephrine (NE) concentration employing a Norepinephrine Enzyme Immunoassay kit (Labor Diagnostika Nord KG, Nordhorn, Germany), based on the manufacturer’s instructions and performed in duplicates for each sample.

### 2.3. Proteomics Analysis

(1) Sample preparation for mass spectrometry

The protein concentration was estimated in each sample using a BCA Protein Assay Kit (Pierce). An amount of 100 μg of protein from each sample was diluted to a 100 μL volume with 100 mM of ammonium bicarbonate (ambic). Proteins were reduced with 5 μL of 200 mM tris(2-carboxyethyl) phosphine (TCEP) (1 h incubation, 55 °C) and alkylated with 5 μL of 375 mM iodoacetamide (IAA) (30 min incubation in the dark, room temperature). The reduced and alkylated proteins were purified with acetone precipitation at −20 °C overnight. The next day, protein precipitates were collected by centrifugation at 8000× *g* for 10 min at 4 °C and pellets were briefly air-dried and resuspended in 100 μL of 50 mM ambic. The protein digestion was carried out using 2.5 μg of trypsin per sample (16 h incubation, 37 °C). The samples were dried out using a speed vacuum and then desalted with C18 spin columns (Pierce). Clean peptides were dried out again with a speed vacuum and resuspended in 100 mM TEAB. The TMT label reagents were equilibrated at room temperature for 10–15 min with the lid sealed, followed by the addition of 42 μL of anhydrous acetonitrile to each of the 0.8 mg TMT label reagents, which was vortexed briefly to make sure the reagents were fully dissolved. The peptides were labeled by adding 42 μL of TMT label reagent to every 100 μL of resuspended peptides and incubated for 1 h at room temperature. Quenching of the reaction was then performed by adding 8 μL of quench buffer to the reaction and incubating for 15 min at room temperature. All labeled peptide samples from the two treatment groups were combined for comparison with the experimental sets containing different tags. The pooled peptide sample was then desalted using high pH reverse-phase HPLC, and the eluted peptides were collected and dried using a speed vac. The dried peptides were finally dissolved in 40 μL 0.1% (*v*/*v*) aqueous FA and analyzed using a high-resolution mass spectrometry nano-LC-MS/MS Orbitrap Exploris 480 coupled with a UltiMate 3000 HPLC system (Thermo Scientific, Waltham, MA, USA).

(2) LC-MS/MS

An amount of 1.5 μg of each sample was loaded onto a trap column Acclaim PepMap 100 (75 μm × 2 cm C18 LC Columns, Thermo Fisher Scientific) at a flow rate of 4 μL/min, and then separated with a Thermo RSLC Ultimate 3000 (Thermo Fisher Scientific) on a Thermo Easy-Spray PepMap RSLC C18 column (75 μm × 50 cm C-18 2 μm, Thermo Fisher Scientific, Waltham, MA, USA) at a flow rate 0.3 μL/min and 50 °C, with a step gradient of 9–25% solvent B (0.1% FA in 80% acetonitrile) for 10–20 min and 25–40% solvent B for 15–120 min, with a 155 min total run time. The MS scan was completed using a detector: Orbitrap resolution 120,000; scan range 350–1800 m/z; RF lens 30%; AGC target 4.0 × 10^5^; maximum injection time 100 ms. The most intense ions with charge states of 2–6 isolated in 3 s cycles were selected in the MS scan for further fragmentation. The MS2 scan parameters set were: activation HCD with 35% normalized collision energy, detected at a mass resolution of 30,000, the AGC target for MS/MS was set at 5.0 × 10^4^, and the ion filling time was set to 60 ms.

(3) Data Analysis

Protein identification was performed by searching MS/MS data against the UniProt database (Rat) in Proteome Discoverer (Thermo Fisher Sci, Waltham, MA, USA, vs. 3.0.), assuming the digestion enzyme trypsin and the 10plex TMT reporter ion MS^2^ mode as the searching mode. The parameters for Sequest HT were set as follows: enzyme: trypsin, max missed cleavage: 2, precursor mass tolerance: 10 ppm, peptide tolerance: ±0.02 Da, fixed modifications: carbamidomethyl (C); dynamic modifications: oxidation (M), acetyl (N-term) and TMT (K)—229.163 Da. The parameters for the precursor ions quantifier was set as follows: peptides to use unique + razor, precursor abundance based on intensity; normalization mode: total peptide amount; scaling mode: on all average.

### 2.4. Bioinformatics Analysis

Proteins were recognized as differentially expressed if the *p*-value of the *t*-test was ≤0.05 and the absolute fold change was ≥1.5. A gene ontology (GO) analysis of the differentially expressed proteins (DEPs) was conducted using the Cytoscape plug-in, ClueGO [17]. Biological processes and molecular functions were included for the GO enrichment analysis. A canonical pathway analysis was performed using Ingenuity Pathway Analysis (IPA) software (Ingenuity Systems, Redwood City, CA, USA, www.ingenuity.com, accessed on 8 August 2023) by comparing the DEPs against known canonical pathways (signaling and metabolic) within the IPA knowledgebase. For further analysis, enriched pathways with a Benjamini–Hochberg false discovery rate (FDR) *p*-value ≤ 0.05 were considered.

### 2.5. Western Blotting Analysis

Protein isolation from the LV myocardium and brain cortex of saline- and GBP- treated animals was isolated using RIPA buffer (50 mM TrisHCl, 195 mM NaCl, 2 mM EDTA, 1% NP-40, 0.1% SDS) with 1% protease inhibitor cocktail (ab65621 Abcam, Cambridge, UK) and then centrifuged at 20,000× *g* × 20 min at 4 °C. The protein concentration was estimated using the Pierce bicinchoninic acid assay (Thermo Fisher Scientific, Waltham, MA, USA). An amount of 10 μg and 5 μg of proteins from the LV myocardium and brain cortex of each animal were loaded separately onto a 10% Bis-Tris wells (Invitrogen, Waltham, MA, USA) under reducing conditions, then transferred to a nitrocellulose membrane using iBlot2 (Invitrogen). Post transfer, the membranes were stained with Ponceau S stain (Thermo Fisher Scientific, Waltham, MA, USA) to assess for equal protein loading detection and quantification. The membranes were treated on a rocker for 1 h at room temperature with 5% fat free milk to block nonspecific antibody binding. After blocking, the membranes were incubated overnight at 4 °C with 1:1000 calmodulin (CaM) antibody (A4885; ABclonal, Woburn, MA, USA). The next day, the membranes were washed and treated with 1:2500 HRP conjugated anti-rabbit IgG for 1 h followed by additional washes. Blots were developed with 1:1 solution of Radiance Chemiluminescent Substrate and Luminol/Enhancer (Azure Biosystems, Dublin, CA, USA). G:BOX CHEMI XRQ imaging system (Syngene, Frederick, MD, USA) was used to visualize the blots, and the images acquired were quantified using the Image-J software version 1.52a.

### 2.6. Statistical Analyses

All data are expressed as mean ± SD. Student’s *t*-test was used to compare the difference between the 2 groups, with the aid of Prism 8 software. A *p* value of <0.05 was taken as indicative of statistical significance.

## 3. Results

### 3.1. Effect of Acute Bolus Intravenous Injection of GBP on Cardiovascular Function

First, we determined the acute responses of BP, HR, and LV hemodynamics to an intravenous injection of GBP (50 mg/kg; 100 mg/mL × 0.1 mL in 3 min). The results are shown in Figure 1, representative of an original recording and the Table 1), the combined summary data.

The recording shows that GBP administration evoked a transient and slight suppression of most hemodynamic parameters, followed by a prolonged and significant inhibition, showing a biphasic effect. Since the *i.v.* catheter tip was placed into the superior vena cava, we believe the inhibitory effect of the 1st phase was induced directly by the outflow of GBP solution from the catheter tip going through the heart chamber and the vessel system. Accordingly, when the injection was completed (indicated by the green arrow), this effect disappears. At approximately 7 min post injection (indicated by the blue arrow) when the GBP solution was completely mixed with circulating blood and the GBP concentration in the blood reached a steady state, we observed the inhibitory effect of the second phase, which lasted at least 60 min with a maximal effect appearing at 30 min post injection (indicated by the yellow arrow). Since the half-life of GBP is 5–7 h, we speculate that *i.v.* GBP-induced cardiovascular inhibition could potentially last longer than one hour.

Table 1 shows that BP, HR, and most LV hemodynamic parameters are significantly reduced after GBP treatment (the values were calculated at 30 min post injection indicated by the yellow arrow) as compared with the baseline (before GBP treatment). The decreased systolic pressure (SP) and pulse pressure (PP) represent impaired cardiac function and reduced cardiac output, whereas the decreased diastolic pressure suggests vasodilation. The lower maximal and minimal dP/dt values suggest depressed myocardial contractility and impaired LV relaxation, which are further confirmed by a significantly decreased IRP-AdP/dt, the slope of a straight line fit to the pressure over the isovolumic relaxation period, in addition to a significantly increased Tau, the exponential time constant of relaxation. However, there was no significant difference in LVEDP between pre- and post-GBP, suggesting that acute GBP-induced depression of cardiac function did not develop into heart failure. It should be noted that an *i.v.* injection of 0.1 mL of saline (vehicle) did not evoke significant changes in BP, HR, and LV hemodynamics over the same time period as rats that underwent GBP injections (data not shown).

### 3.2. Effect of Chronic i.p. Treatment of GBP on Cardiovascular Function

We determined the chronic effects of GBP administration (*i.p.* 100 mg/kg/day for seven consecutive days) on cardiovascular function compared with saline as the control vehicle. The measurements were carried out 6 h after last GBP administration on day 7. The results are shown in Figure 2, in which the left panel is representative of the original recordings and the right panel shows the combined data (Table 2). GBP-treated rats exhibited significantly lower BP, HR, and LV hemodynamics compared to the rats receiving *i.p.* saline, suggesting that the inhibitory effects of chronic GBP treatment on cardiovascular function was similar to those induced by acute GBP-treatment. However, some parameters, which were significantly reduced in the acute GBP-treated rats (Table 1), did not display a significant change in chronic GBP-treated rats (Table 2), such as PP, dP/dt_min_, IRP-Average dP/dt, Tau, DD, and CD, suggesting a different cardiovascular response to acute- and chronic- GBP administration. However, this difference could also be attributed to the different time points the two groups chose for calculating the parameters, since we evaluated the acute response after 30 min post GBP treatment (Table 1) but evaluated the chronic response at 6 h after GBP administration (Table 2).

### 3.3. Plasma NE Concentration

To determine whether the sympathetic system is involved in GBP-induced cardiovascular dysfunction, we next measured the plasma NE concentration in the rats receiving chronic GPB treatment. No significant differences in plasma NE concentration were observed between the two groups (Figure 3), which suggests that the 7-day *i.p.* treatment of GBP in a dose of 100 mg/kg does not change circulating catecholamine and sympathetic tone.

### 3.4. Proteomics and Bioinformatics Analyses

To explore the molecular mechanisms underlying chronic GBP treatment-induced cardiac dysfunction, we employed Tandem Mass Tag (TMT) multiplexing coupled to liquid chromatography-mass spectrometry based quantitative proteomics to analyze LV myocardium. A total of 2262 proteins with a minimum of two plus unique peptides were identified, of which 109 proteins were found to be differentially expressed between the groups (Appendix A). Panel A of Figure 4 shows the principal component analysis (PCA), which represents an overall separation and reproducibility amongst the biological replicates that occurred between GBP- and saline-treated rats with 65.2% of variation. Panel B of Figure 4 is a volcano plot indicating significantly upregulated (red dots, n = 45) or downregulated (blue dots, n = 64) proteins in GBP rats compared to saline rats. Panel C of Figure 4 is a heatmap analysis of the top 121 proteins, revealing a favorable consistency of protein profiles between the groups. Next, using ClueGO, a user friendly cytoscape plug in, we analyzed both the molecular and biological processes enriched within the differentially expressed proteins. Amongst the molecular functions, we found enrichment of ribonucleoprotein complex (11.8%), heat shock binding (10.75%), and RNA helicase activity (10.75%). Similarly, in the biological functions, we found enrichment in the translation regulator activity (31.34%), followed by nucleic acid transport (14.93%), and negative regulation of chromatin silencing (11.94%). These data are presented in Figure 5.

### 3.5. Western Blotting Analysis

High throughput studies generate vast amounts of data and initially identify potential hits, thus there is the need for further validation. One such protein hit we identified from the proteomics screen was calmodulin (CaM), which was +1.52 fold upregulated in the GBP group. To further validate its upregulation, we performed a western blot on the heart lysates from the two groups. As seen in Figure 6, CaM expression was significantly elevated (+1.77 fold) in the heart lysates of the GBP-treated animals, thus validating our proteomics screen data. To further ascertain whether the upregulation of CaM is heart specific, we also examined its expression in the brain cortical lysates of the two groups. Notably, there was no change in CaM expression in the brain lysates of the GBP animals. This suggests that the 7-day *i.p*. treatment of GBP with a dose of 100 mg/kg/day significantly upregulates CaM in the myocardium specifically, which may contribute to the subsequent cardiac dysfunction.

## 4. Discussion

Although GBP has long been used in a neurosetting, its cardiovascular side effects have recently attracted attention in human medicine, veterinary medicine, and basic medical science areas [8,18]. However, the exact actions and underlying mechanisms of GBP on the heart, blood vessels, and autonomic regulation remain to be completely elucidated. The novel findings of the present study are that both acute and chronic systemic administration of GBP in rats evoked a significant inhibition of cardiac function. This included bradycardia, depressed myocardial contractility, and LV systolic/diastolic dysfunction, which were accompanied by a decrease in BP without a change in plasma norepinephrine concentration. These data imply a direct effect of GBP on the heart. In addition, when using mass spectrometry-based proteomics, we identified 109 differentially expressed proteins in heart samples of rats receiving chronic GBP treatment. Bioinformatics analysis further suggests that these proteins are involved in calcium signaling pathways, cholesterol metabolism, galactose metabolism, and other biological processes associated with cardiac function. One of the most important pathways that we identified using proteomics and validated by western blot in this study was CaM, a highly conserved protein that senses intracellular Ca^2+^ and relays signals to various Ca^2+^-sensitive enzymes, ion channels, and other proteins [19]. We found that CaM was markedly upregulated in the myocardium, but not in the brain cortices of the rats receiving chronic GBP treatment. This finding strongly suggests an association of the observed cardiac dysfunction with impaired regulation of cardiomyocyte Ca^2+^ signaling. To the best of our knowledge, this is the first study to provide physiological and molecular evidence suggesting a direct effect of GBP on cardiac function and protein profiles by combining whole animal experiments with omics techniques and molecular validation. Consistent with our findings in normal rats, a recent study by Allen et al. [20] demonstrated that a single dose of oral GBP in healthy cats produced a modest decrease in LV systolic function, displaying a significant decrease in two-dimensional fractional shortening and a significant increase in both LV internal diameter in systole and left atrial volume.

The effects of GBP on cardiovascular function in normotensive and hypertensive rats has been investigated recently by several studies. In conscious spontaneously hypertensive rats (SHR) and Wistar-Kyoto (WKY) rats, Behuliak et al. [15] found that acute intravenous injection of GBP significantly reduced BP and HR, with greater effects in SHR than WKY. They further demonstrated that the hypotension and bradycardia were mediated by attenuating sympathetic nerve transmission and modulating the arterial baroreceptor-HR reflex. To explore the central mechanisms of GBP-evoked cardiovascular depression, Chen et al. [14] found that microinjection of GBP into the nucleus tractus solitarii (NTS) of SHR rats induced a dose-dependent decrease in BP and HR, which was completely abolished by pretreatment with L-NAME, a NOS inhibitor. These data suggest a critical role of GBP in central autonomic regulation via a NO-dependent pathway. In contrast, to explore the peripheral mechanisms of GBP-induced edema and acute heart failure, Largeau et al. [16] utilized pressure arteriography and whole-cell patch clamp to study the direct effects of GBP on vasomotor capacity of third-order mesenteric arteries and L-type calcium current in LV cardiomyocytes isolated from adult male Wistar rats. They found that GBP evoked a marked decrease in myogenic tone and weak vasodilation but there were no effects on cellular Ca^2+^ current. These data are the first pieces of evidence demonstrating how GBP can modulate cardiovascular function through a direct effect on non-nervous tissues. In the present study, by simultaneously measuring BP, HR, and LV hemodynamics in anesthetized rats, we found an immediate hypotension, bradycardia, and impaired LV contractility/relaxation in response to intravenous injection of GBP, further confirming the direct effects of GBP on the heart and blood vessels.

There are no clinical studies to have investigated the side effects or toxicity of GBP on cardiovascular function, however, several case reports describe adverse cardiovascular events induced by GBP or its structurally similar compound, pregabalin. These include the development of new onset congestive heart failure [10,11], decompensation of pre-existing CHF [21,22,23], cardiac conduction disturbances [24,25], atrial fibrillation [9], and hypotension [13,26]. In an analysis of a large-scale nation-wide database of de-identified patients (TriNetZ EHRs), Pan et al. [8] recently found a significant increase in the risk of cardiovascular diseases, including heart failure, myocardial infarction, peripheral vascular disease, stroke, deep venous thrombosis, and pulmonary embolism in patients with diabetic neuropathy following long-term use of GBP and pregabalin. Furthermore, they found that there were significant associations between short-term (3 month) GBP use and heart failure, myocardial infarction, peripheral vascular disease, deep venous thrombosis, and pulmonary embolism. The data in the present study provides animal-based experimental evidence to support the cardiovascular dysfunction described by these clinical case reports and the high risk of cardiovascular diseases found by the retrospective cohort study.

Although autonomic dysfunction has been proposed to explain the negative influence of GBP on the cardiovascular system [14,15], the exact mechanisms remain to be explored. A decreased peripheral sympathetic nerve transmission [15] and suppressed central sympathetic nerve outflow [14] can explain GBP-induced hypotension and bradycardia, however this not likely the cause of GBP-evoked new onset congestive heart failure [10,11], decompensation of pre-existing heart failure [21,22,23], or an increase in the risk of heart failure [8], since sympatho-inhibition is generally believed to ameliorate, rather than exacerbate, the cardiac dysfunction in the heart failure state. Based on our data showing acute cardiac dysfunction by GBP and marked upregulation of CaM in the myocardium of rats receiving chronic GBP, we hypothesize that GBP-induced cardiovascular pathological consequences may be attributed to a direct effect of GBP on myocardial contractility induced by abnormal Ca^2+^ signaling in cardiomyocytes.

Despite its structural similarity to GABA, GBP cannot bind to GABA receptors or affect the neuronal uptake or degradation of GABA, suggesting that the effects of GBP are not mediated by interacting directly with GABA receptors or with high-affinity Na^+^-dependent GABA transporters [27]. Instead, GBP directly inhibits the voltage-gated calcium channels (VGCCs) of neurons, leading to a reduction in presynaptic Ca^2+^ influx, a decrease in the release of excitatory neurotransmitters, and a relief of neuropathic pain [28]. VGCCs are composed of a complex, consisting of an α1 and associated β and α2δ subunits, to which the α1 subunit constitutes the channel pore and allows calcium influx from the extracellular space into the cells. The β and α2δ subunits support channel trafficking and tune the kinetic properties of Ca^2+^ currents [29,30]. By binding to the α2δ subunits of the VGCC complex, GBP blocks neuronal Ca^2+^ influx, exerting its analgesic efficacy [28,31]. In addition to the nervous system, VGCCs are also found in other excitable tissues, such as cardiac muscle, skeletal muscle, smooth muscle, and endocrine cells, playing a critical role in the regulation of muscle contraction and hormone secretion [30]. In the present study, we propose that the cardiac dysfunction observed in rats (and in other studies with cats) [20] is related to a blockade by GBP on cardiomyocytes L-type Ca^2+^ current by binding to the α2δ subunit of the Ca_V_1.2 channel, a cardiac subtype of VGCCs. Moreover, by employing a [^3^H]GBP binding assay and α2δ-null mice, Fuller-Bicer et al. [32] demonstrated a binding capacity of GBP with VGCCs in cardiomyocytes. Furthermore, they found that α2δ-deficient hearts display a significantly lower basal contractility and relaxation (+dP/dt and -dP/dt) as compared with the wild type [32]. Actually, VGCCs participate in and are critical for almost all aspects of cardiac physiology, pathology, and pharmacology, through regulating extracellular Ca^2+^ influx and subsequent intracellular signaling transduction [33,34].

Cardiomyocyte VGCCs, the Ca_V_1.2 channel subunit, mediates excitation-contraction coupling (ECC), a highly coordinated process whereby membrane depolarization leads to contraction [35]. During the period of action potential generation, small amounts of extracellular Ca^2+^ ions enter the cytosol via the Ca_V_1.2 and trigger the release of massive amounts of intracellular Ca^2+^ from the sarcoplasmic reticulum (SR) into the cytosol via the ryanodine receptor (RyR, a Ca^2+^ release channels resided in SR), known as Ca^2+^-induced Ca^2+^ release (CICR) [36]. These Ca^2+^ ions released from SR bind to troponin-C and enable actin-myosin cross-bridging and sliding of the myofilaments, resulting in sarcomere shortening and myocardial contraction [37]. CICR is a complex process critical for ECC, not only involving in Ca_V_1.2 and RyR but also CaM. CaM is a Ca^2+^ binding protein, which crosstalks with both Ca_V_1.2 and RyR, forming a triangular interaction to regulate Ca^2+^ behavior and kinetics [38]. On one hand, the functional communications between the RyR and Ca_V_1.2 are believed to be reciprocal. Ca_V_1.2 opens RyR, defined as orthograde signaling, and RyR can prevent Ca_V_1.2 inactivation, defined as retrograde signaling [39,40]. On the other hand, CaM in both Ca^2+^-bound and Ca^2+^-free states can bind and regulate both RyR and Ca_V_1.2. RyR activity is increased by CaM at low Ca^2+^ concentrations (nM; Ca^2+^-free CaM) but inhibited by CaM at high Ca^2+^ concentrations (μM; Ca^2+^-bound CaM), whereas Ca_V_1.2 can be inactivated and facilitated by CaM via its binding to channel’s C-terminal domain as a Ca^2+^ sensor [38]. One of the important findings from the present study is a marked upregulation of CaM in the hearts of rats receiving the 7-day GBP treatment, suggesting an impact of GBP on cardiomyocyte Ca^2+^ signaling which we believe underlies the subsequent cardiac dysfunction. Indeed, it has been found that hyper-activated Ca^2+^/CaM signaling contributed to pathology in several cardiac diseases, including heart failure, cardiac hypertrophy, and arrhythmia [41,42]. Although the exact mechanisms by which GBP upregulates CaM and the resulting functional consequences by the upregulated CaM remain unclear, though our data does suggest the involvement of intracellular Ca^2+^ signaling dysfunction in GBP-induced cardiac side effects and will open novel avenues for treating in the future.

In conclusion, the present study revealed that both acute and chronic treatments with GBP in normal adult rats had a significant adverse effect on global cardiovascular function, including hypotension, bradycardia, and reduced LV systolic/diastolic function. This is the first study to provide direct evidence demonstrating inhibition of GBP on cardiac contractility. Given the observed substantial increase in myocardial CaM protein expression in the GBP-treated rats, we speculate that a disruption in Ca^2+^ signaling pathways within cardiac myocytes plays a pivotal role in these effects.

## Figures and Tables

**Figure 1 cells-12-02705-f001:**
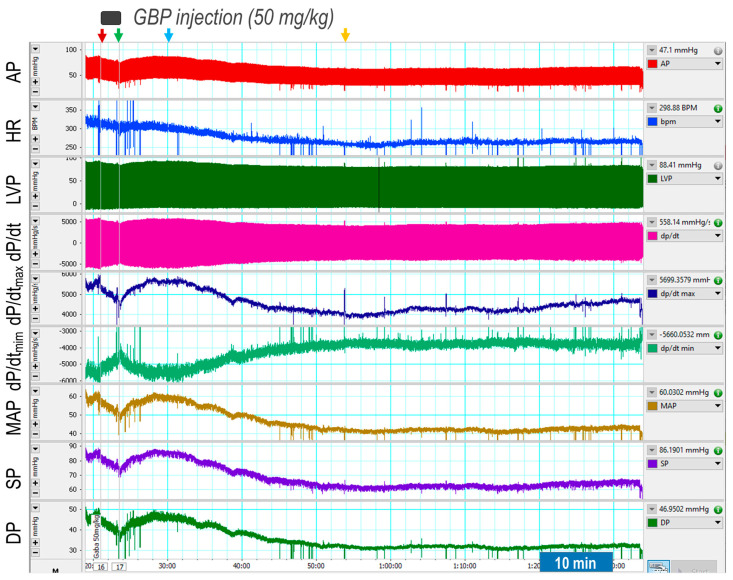
Representative of an original recording showing effects of a bolus *i.v.* injection of GBP on cardiovascular function. The red and green arrows indicate the start and end of a GBP injection, respectively. The yellow arrow indicates the maximal effect of GBP.

**Figure 2 cells-12-02705-f002:**
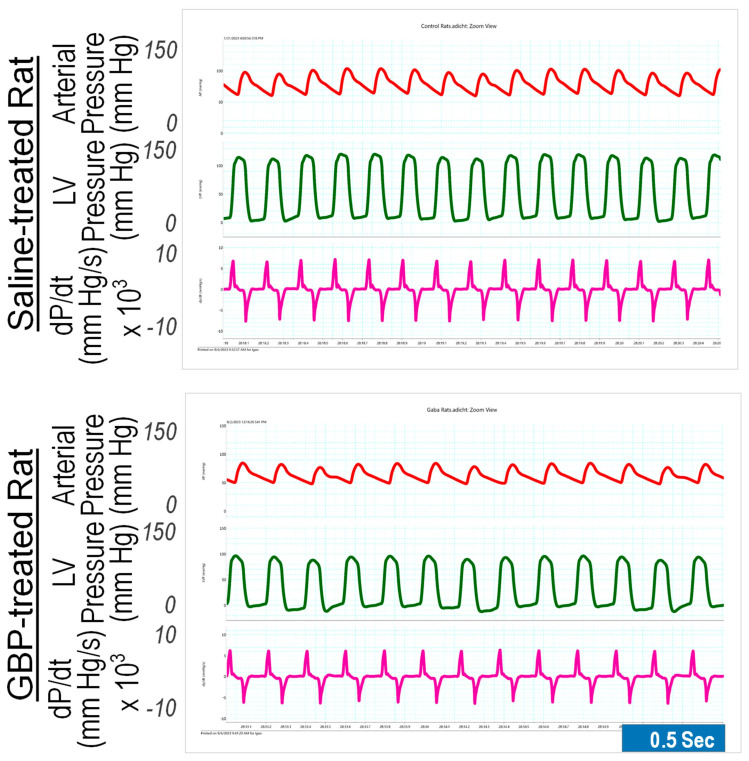
Representative of original recordings showing effects of the 7-day *i.p.* injection treatment of GBP on cardiovascular function. Top panel: a saline-treated rat; Bottom panel: a GBP-treated rat.

**Figure 3 cells-12-02705-f003:**
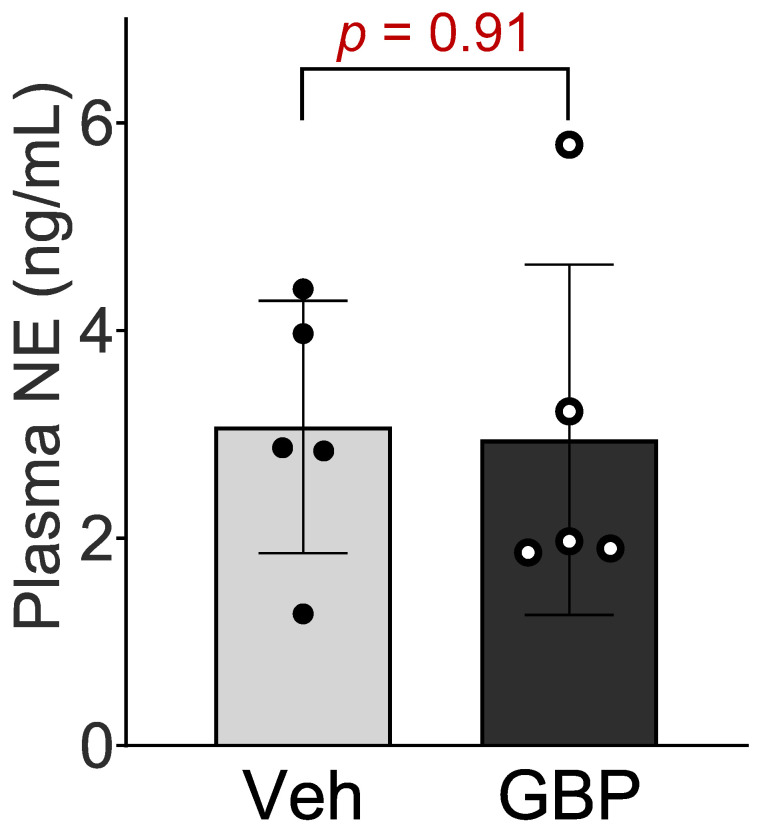
Plasma norepinephrine concentration of the rats receiving the 7-day saline or GBP treatment. n = 5/group. NE: norepinephrine; Veh: vehicle; GBP: gabapentin.

**Figure 4 cells-12-02705-f004:**
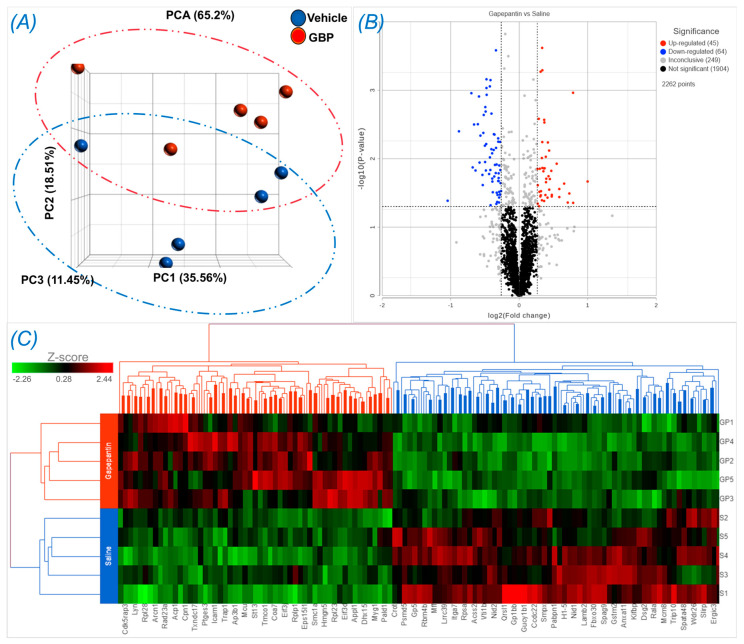
Differentially expressed proteins in the LV myocardium of rats receiving GBP or saline treatment for 7 days. (**A**) Principal component analysis (PCA) showing a separation occurred between the groups. (**B**) Volcano plot showing the log2 fold change plotted against the −log10 *p* value; (**C**) Heatmap showing the log2 of the top 121 differentially expressed proteins with 55 up- and 66 down-regulation. n = 5/group.

**Figure 5 cells-12-02705-f005:**
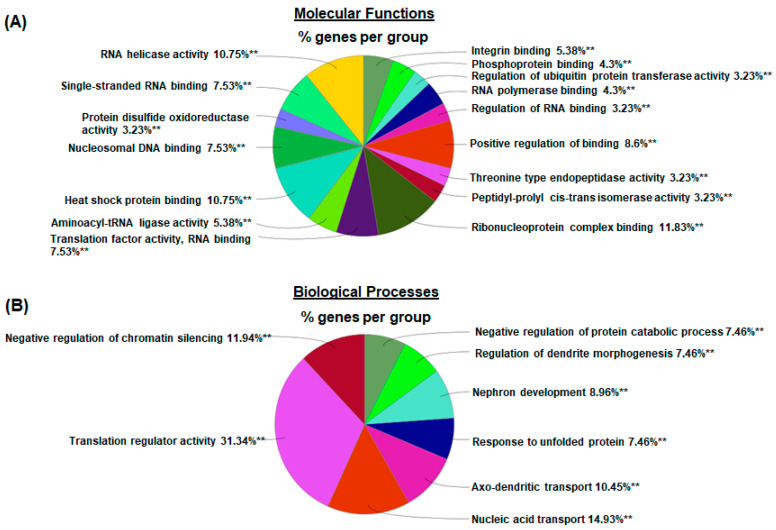
Pie chart representing the distribution of differentially expressed proteins identified in the LV myocardium of GBP-rats compared to saline-rats, according to the proteins’ molecular functions (**A**) and biological processes (**B**) by using ClueGO analysis. ** *p* < 0.001 as compared with saline groups, n = 5/groups.

**Figure 6 cells-12-02705-f006:**
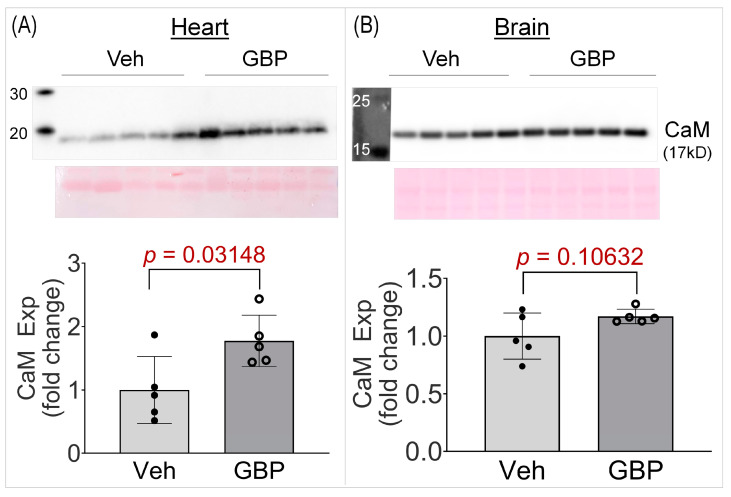
Western blot data showing CaM expression in the LV myocardium (**A**) and brain cortex (**B**) of rats receiving the 7-day Veh or GBP treatment. n = 5/group. Veh: vehicle (saline); GBP: gabapentin; CaM: calmodulin; Exp: expression.

**Table 1 cells-12-02705-t001:** Combined data showing effects of a bolus *i.v.* injection of GBP on cardiovascular function.

Parameters	Before GBP (Baseline)	After GBP (at 30 min)
MAP (mmHg)	90.57 ± 21.27	52.88 ± 6.65 **
SP (mmHg)	111.46 ± 22.46	68.88 ± 7.37 **
DP (mmHg)	73.49 ± 20.68	40.35 ± 6.47 *
PP (mmHg)	37.96 ± 2.02	28.52 ± 3.23 **
HR (bpm)	369.22 ± 53.71	299.87 ± 31.68 *
T-to-P (s)	0.035 ± 0.005	0.047 ± 0.006 *
LVP_max_ (mmHg)	118.01 ± 23.21	85.69 ± 4.64 *
LVP_min_ (mmHg)	−1.88 ± 4.63	−2.04 ± 4.81 ^ns^
LVEDP (mmHg)	6.03 ± 4.46	4.91 ± 3.99 ^ns^
dP/dt_max_ (mmHg/s)	7746.76 ± 2022.69	4919.22 ± 919.05 *
dP/dt_min_ (mmHg/s)	−8822.36 ± 2783.16	−4662.46 ± 677.01 *
CI (1/s)	112.44 ± 12.58	108.46 ± 15.66 ^ns^
PTI (mmHg.s)	7.52 ± 0.94	6.01 ± 0.15 ^ns^
IRP-AdP/dt (mmHg/s)	−4610.35 ± 936.66	−2762.92 ± 262.55 *
Tau (s)	0.0129 ± 0.0025	0.0153 ± 0.0022 *
SD (s)	0.081 ± 0.007	0.091 ± 0.009 *
DD (s)	0.085 ± 0.017	0.111 ± 0.015 *
CD (s)	0.165 ± 0.024	0.201 ± 0.024 *

Data show as Mean ± SD; * *p* < 0.05, and ** *p* < 0.01, compared to before GBP. n = 4. ns: no statistical significance. CI: Contractility Index; PTI: Pressure Time Index; IRP-AdP/dt: Isovolumetric relaxation average dP/dt. SD, DD, and CD: systolic, diastolic, and cycle duration.

**Table 2 cells-12-02705-t002:** Combined data showing effects of the 7-day *i.p.* injection treatment of GBP on cardiovascular function.

Parameters	Saline-Rats	GBP-Rats
MAP (mmHg)	93.49 ± 3.28	73.96 ± 11.83 **
SP (mmHg)	114.33 ± 3.24	94.01 ± 12.93 **
DP (mmHg)	76.47 ± 3.79	58.02 ± 11.22 **
PP (mmHg)	37.86 ± 3.22	35.99 ± 2.38 ^ns^
HR (bpm)	383.15 ± 26.24	317.89 ± 12.12 **
T-to-P (s)	0.032 ± 0.001	0.042 ± 0.004 ***
LVP_max_ (mmHg)	120.68 ± 15.91	93.06 ± 8.48 **
LVP_min_ (mmHg)	3.26 ± 8.65	−3.49 ± 1.16 ^ns^
LVEDP (mmHg)	7.77 ± 6.73	5.18 ± 2.62 ^ns^
dP/dt_max_ (mmHg/s)	7692.31 ± 1672.35	5527.34 ± 884.97 *
dP/dt_min_ (mmHg/s)	−8822.36 ± 3508.77	−6011.48 ± 776.33 ^ns^
CI (1/s)	112.55 ± 12.02	108.65 ± 3.59 ^ns^
PTI (mmHg.s)	7.75 ± 0.49	6.39 ± 0.56 ^ns^
IRP-AdP/dt (mmHg/s)	−3687.47 ± 1722.11	−3455.91 ± 386.52 ^ns^
Tau (s)	0.0154 ± 0.0045	0.0147 ± 0.0008 ^ns^
SD (s)	0.079 ± 0.005	0.086 ± 0.002 *
DD (s)	0.089 ± 0.019	0.107 ± 0.007 ^ns^
CD (s)	0.168 ± 0.024	0.193 ± 0.007 ^ns^

Data presented as Mean ± SD; * *p* < 0.05, ** *p* < 0.01, and *** *p* < 0.001, compared with the saline-treated group. n = 5/group. Please refer to Table 1 for the meaning of the abbreviations.

## Data Availability

The data are contained within the article and Appendix A.

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
