# Peer review of "Effects of Acute and Chronic Gabapentin Treatment on Cardiovascular Function of Rats"

_cells, 2023, doi:10.3390/cells12232705_

Round 1

Reviewer 1 Report

Comments and Suggestions for Authors

Although known through previous literature on Gabapentin mediated reduction in Blood Pressure and Heart Rate, Pendyala et al explored the molecular mechanisms underlying GBP-induced cardiac dysfunction and the results are well represented and convincing.The manuscript can be accepted in current form.

Author Response

We greatly appreciate your recognition of our work and your recommendation to accept our manuscript for publication.

Reviewer 2 Report

Comments and Suggestions for Authors

- Simple Summary is missing - please add it to the manuscript

- Materials and Methods: it is unclear how many animals were used in the acute phase of the experiment; in line 93 Authors mention that 8 animals were included in Group 1, while in line 220 (Figure 1 legend) it is stated that n=4/group; if two groups were studied, it is unclear what was the difference between the groups. If only one group consisting of 4 animals was studied, it should be corrected in Mat&Met section

- Materials and Methods: what was the method to calculate the sample size for each group?

- Results: please unify the legends for Figure 1 & Table 1 and Figure 2 & Table 2: either describe table as a part of a figure (like in Figure & Table 1) or as separate objects (as in Figure & Table 2)

- Figure 3: please explain "NE", "GBP" and "Veh" in the legend

- Lines 273-281 is not a result; it should be a figure legend with only a summary mentioned in the results section

- Figure 5 should show a comparison between GBP- and saline-treated rats, while it shows (probably?) only GBP-treated rats? The results obtained from saline-treated group should also be presented in the chart; moreover, I suggest changing the chart type from pie chart to column or stacked column chart to visualise the differences

- Figure 6: please explain the abbreviations used in the figure
